# Molecular and Biochemical Evidence of the Toxic Effects of Terbuthylazine and Malathion in Zebrafish

**DOI:** 10.3390/ani13061029

**Published:** 2023-03-11

**Authors:** Ihab Khatib, Oksana Horyn, Oksana Bodnar, Oleh Lushchak, Piotr Rychter, Halina Falfushynska

**Affiliations:** 1Chemical-Biological Faculty, Ternopil Volodymyr Hnatiuk National Pedagogical University, 46027 Ternopil, Ukraine; 2Department of Biochemistry and Biotechnology, Vasyl Stefanyk Precarpathian National University, 76018 Ivano-Frankivsk, Ukraine; 3Faculty of Science & Technology, Jan Dlugosz University in Czestochowa, Armii Krajowej 13/15, 42-200 Czestochowa, Poland; 4Department of Marine Biology, Institute for Biological Sciences, University of Rostock, 18059 Rostock, Germany; 5Department of the Electrical Engineering, Mechanical Engineering and Industrial Engineering, University for Applied Sciences-Anhalt, 06366 Köthen, Germany

**Keywords:** *Danio rerio*, chloro-s-triazine herbicide, organophosphate insecticide, oxidative stress, apoptosis, immune toxicity, cytotoxicity

## Abstract

**Simple Summary:**

Due to the global increase in pesticide applications, aquatic animals are constantly subjected to their action in natural reservoirs. Terbuthylazine and malathion, two commonly detected pesticides in water and soil samples elsewhere, have been shown to have a significant negative impact on zebrafish, even at environmentally realistic concentrations. They can induce oxidative stress, mitochondrial and lysosomal destabilization, leading to immune toxicity, cytotoxicity, and DNA damage. All of these adverse outcomes might severely impact the health status of fish before being translated into an effect on the population. The present study should emphasize the importance of paying attention to pesticide traces in the environment, which can have devastating effects on biota even at low concentrations.

**Abstract:**

Our research sought to determine the molecular and biochemical effects of environmentally relevant exposure to commonly used chloro-s-triazine herbicide terbuthylazine and organophosphate insecticide malathion on zebrafish. To this aim, mature zebrafish were exposed to 2 and 30 µg L^−1^ terbuthylazine and 5 and 50 µg L^−1^ malathion alone and in combination for 14 days. Aside from the accumulation of TBARS and protein carbonyls, a decrease in antioxidants and succinate dehydrogenase activity, an increase in oxidized glutathione, and enhanced apoptosis via Caspase-3 and BAX overexpression were observed. Furthermore, terbuthylazine and malathion induced mitochondrial swelling (up to 210% after single exposure and up to 470% after co-exposure) and lactate dehydrogenase leakage (up to 268% after single exposure and up to 570% after co-exposure) in a concentration-dependent manner. Significant upregulation of ubiquitin expression and increased cathepsin D activity were characteristics that appeared only upon terbuthylazine exposure, whereas the induction of IgM was identified as the specific characteristic of malathion toxicity. Meanwhile, no alterations in the zebrafish hypothalamic-pituitary-thyroid axis was observed. Co-exposure increased the adverse effects of individual pesticides on zebrafish. This study should improve the understanding of the mechanisms of pesticide toxicity that lead to fish impairment and biodiversity decline.

## 1. Introduction

Despite the steady increase in human acute poisoning cases (nearly 740,000 annually in 2020 [1]), and prominent harm to the environment, the total reported sales of pesticides across the world are continuously increasing due to their importance in the protection of plants from pests, weeds, or diseases and humans from vector-borne diseases [2]. Following Worldometer data, approximately 4 million tonnes of pesticides are utilized annually [3].

In addition to fungicides and insecticides, herbicides make up around 95% of all pesticides in European countries [4]. The extensive use of these chemicals might lead to their appearance in different spots, including soil, surface waters, and ground waters. It is no surprise that the amount of agricultural land at risk for pesticide pollution has been steadily rising [5,6]. Almost two-thirds of global agricultural land is now polluted with more than one pesticide, and a third of those areas are at high risk [7]. It has been reported that the bulk of rural water bodies do not meet the EU Water Framework Directive’s standards for high ecological conditions, especially in the Scandinavian and Baltic countries (from 6% to 19%) [8]. Nevertheless, the use of pesticides is predicted to significantly increase in the future.

Terbuthylazine (6-chloro-N2-ethyl-N4-tert-butyl-1,3,5-triazine-2,4-diamine) has lower water solubility and higher soil retention than atrazine (6-chloro-N2-ethyl-N4-isopropyl-1,3,5-triazine-2,4-diamine), and is therefore considered less toxic, reducing aquatic biocontamination [9]. However, from a chemical point of view, these substances belong to the class of nitrogen-containing heterocycles (s-triazines). The difference in structure between both pesticides comes from the tert-butyl group in terbuthylazine, which replaced the more hydrophilic isopropyl group in atrazine. Therefore, it is expected that such a substance should distribute slower in water and, at the same time, should not be as toxic as atrazine due to its lower solubility in water.

Although there are still doubts about whether atrazine should be banned or not, the last decision of the EPA [10] allows for the use of this pesticide. Keeping in mind that atrazine, as a member of the triazine class, is extremely toxic to living organisms, its analogue terbuthylazine should be closely monitored to avoid irreversible changes in the environment at each level of the food chain. Only early reaction and early response may limit the biomagnification process within a trophic level. Analysis of WHO, EPA, and ECHA reports devoted to examined pesticides indicate that there is still missing information devoted to the toxicity effect on *Danio rerio* [11,12,13,14]. The obtained results aid in filling these gaps.

Due to the potential for antagonistic or even synergistic effects, contaminants from different chemical groups and their interactions are now a primary focus. Animals frequently exhibit different biochemical responses when exposed to the same pesticides in a mixture of pesticides [15]. This is a worrisome finding because when pesticides enter natural reservoirs, they meet other chemicals, can interact with each other, and aggravate final adverse outcomes due to cocktail effects. To elaborate, it has been recently shown that triazine herbicides at micromolar concentrations deteriorate chlorpyrifos toxicity to *Hyalella azteca*. The highest index of synergistic ratio was obtained for atrazine (SR = 1.42) [16]. Furthermore, chlorpyrifos promoted the expression of IL-6, IL-8, and TNF- in carp tissues, which was significantly enhanced when combined with the triazine herbicide atrazine [17]. Although there is some information regarding aggravation of the health status of animals after co-exposure with organophosphate pesticides, which are the most widely used across the world, and ring-based triazines, there is a lack of information regarding their interaction with terbuthylazine, which has started to substitute for atrazine across the world.

Their movement and relatively lengthy lifespan make fish an effective bioindicator species and a robust model to study biochemical and physiological responses to both short- and long-term toxic effects and various habitat conditions. Additionally, they reflect links between individual responses and adverse outcomes at organismal and population levels. Among others, zebrafish (*Danio rerio*) are one of the most widely-used aquatic animal models for a number of reasons. Firstly, *Danio rerio* is a prominent animal model for toxicological research, given its rapid and precise insights into absorption, distribution, metabolism, and modus operandi of toxins. As in humans, a compound should enter an organism, overcome metabolic processes and cross the blood–brain barriers of zebrafish to affect it. There are numerous findings that assume oxidative stress signs, inflammation, disturbances in neural and immune systems, as well as disruption in hormonal pathways in zebrafish after pesticide exposure [18,19,20,21,22]. Furthermore, zebrafish have an inherent integrative conserved physiology and possess all the main organs and physiological systems of humans, which may not be obvious at first glance. In addition, the *Danio rerio* genome highly resembles that in humans, with 71% of human proteins having orthologues in zebrafish [23].

Therefore, considering the necessity to improve the ecotoxicological assessment of triazine pesticides alone and in combination with widespread organophosphates, particularly considering their affinity for water and potential to mix up in natural reservoirs, our work aimed to evaluate the molecular and biochemical effects of a transient environmental exposure to terbuthylazine and malathion on zebrafish. By applying a multi-level biomarker approach, including molecular, biochemical, and metabolic responses, this study should enhance the understanding of processes leading to individual impairment caused by pesticide pollution.

## 2. Materials and Methods

### 2.1. Experimental Exposures

Mature males of the AB wild-type zebrafish *D. rerio*, aged 8–12 months, were used to test the harmful effects of tetrazine and organophosphate pesticides, both alongside each other and in combination. The Ternopil V. Hnatiuk National Pedagogical University’s animal ethics committee accepted all studies as long as they complied with the standards for the welfare of laboratory animals (Protocol No. 2; August 2020). A total of 315 specimens of zebrafish of the same size (3.4 ± 0.4 cm) and the same mass (0.9 ± 0.2 g) were purchased from a local vendor (Rubkavdoma, Kyiv, UA). The fish were randomly assigned to one control group (not exposed to chemicals) and six experimental groups, each of which was exposed to terbuthylazine, malathion, or mixtures in two different concentrations. At 18 °C, experimental fish were subjected to studied pesticides in the following concentrations for 14 days: (1) terbuthylazine, 2 µg L^−1^ (TL), (2) terbuthylazine, 30 µg L^−1^ (TH), (3) malathion, 5 µg L^−1^ (ML), (4) malathion, 50 µg L^−1^ (MH), (5) terbuthylazine 30 µg L^−1^ and malathion 5 µg L^−1^ (TH + ML), (6) terbuthylazine 2 µg L^−1^ and malathion 50 µg L^−1^ (TL + MH). While the higher tested concentrations represent the worst-case scenarios, such as what might be anticipated close to the agricultural wastewater discharge, the lower of the two experimental concentrations of the studied pesticides is in the range of the reported environmental values in freshwaters (0.3–5 µg L^−1^ for terbuthylazine and 2–18.12 µg L^−1^ for malathion) [24,25,26,27]. The control fish were kept in filtered tap water for 14 days. Three replicate tanks with 15 fish each in 10 L of water were utilized for each experimental treatment. The fish were fed commercial fish flakes (Aquarius, Kharkiv, Ukraine) daily and kept in a 12 h:12 h light:dark cycle. By using the dissolved oxygen meter Eutech DO 6+ (Thermo Fisher Scientific, Waltham, MA, USA) and pH meter MI 150 M (Minsk, Belarus), respectively, the dissolved oxygen concentration and pH were monitored five times per day and kept constant in the range of 7.9–8.2 mg/L for dissolved oxygen concentration and 7.6 for pH. Having been monitored properly by conventional techniques, electrical conductivity, hardness, chloride concentrations, and oxidizability did not exceed the limits for fresh and tap water. To keep the water proper for animal housing and the pesticide levels at declared nominal levels, it was changed every other day. In order to minimize the time between the water change and the preparation of the stock and working solutions of the studied pesticides, we did all of our work just before the water change. Before introducing fish to the exposure tanks, pesticides were administered, and the water was aerated for 10 min to ensure that the pesticides were evenly distributed throughout the system. To prevent stressing out the fish, terbuthylazine and malathion were diluted in fresh water and applied gradually during subsequent water changes. Over the course of the exposures, no fish deaths were observed.

The liver and brain were promptly dissected on ice after the fish were euthanized by causing hypothermic shock in an ice-chilled water bath (5:1 ice to water, 2–4 °C) after exposure. Blood was collected using a micropipette tip and a lateral incision close to the dorsal aorta. For the purpose of isolating serum, the whole blood from five animals (~5–7 μL each) was combined and centrifuged for 10 min at 500× *g*.

To determine the biological traits, the liver and brain were homogenized (1:10 w:v) in 0.1 M phosphate buffer pH 7.4, containing 100 mM KCl, 1 mM EDTA and 0.1 mM phenylmethylsulfonyl fluoride. An aliquot of homogenate was centrifuged at 6000× *g* for 10 min and the supernatant was used for measurements. The amount of protein in the supernatant was determined using a Lowry (1951) protein assay with bovine serum albumin as a standard. The absorbance values were measured on UV/Vis spectrophotometer U-Lab 101UV (ULab, Shanghai, China) and Multiskan™ FC Microplate Photometer (ThermoFisher Scientific, Waltham, MA, USA).

### 2.2. Antioxidants

The total antioxidant capacity (TAC) as the measure of free radicals scavenged by zebrafish hepatocytes was determined in soluble protein fractions based on the rate of decrease in the absorbance of 2,2′-azinobis (3-ethylbenzothiazoline 6-sulfonate)^+^ radical at 734 nm [28]. Trolox solutions were used as standards for calibration. Trolox equivalent antioxidant capacity was presented as nmol Trolox equivalent mg^−1^ proteins.

The activity of catalase (CAT, EC 1.11.1.6) was determined in the soluble protein fraction of liver tissue homogenate (1:10 *w:v*) using hydrogen peroxide as a substrate [29]. The CAT activity was determined at 240 nm for 60 s and calculated using the extinction coefficient, *ε* = 40 M^−1^ cm^−1^.

The glutathione reductase enzymatic recycling protocol utilizing Ellman’s reagent as a chromogen was used to quantify total glutathione (GSH) in liver tissue homogenate [30]. The thionitrobenzoate rate was measured at 412 nm. The samples were pre-treated with 2% 2-vinylpyridine in order to determine the amount of oxidized glutathione (GSSG) [31]. GSSG was used to create standard curve, and the tissue concentration was reported in µmoles per gram.

### 2.3. Oxidative and Nitrosative Stress Markers

According to Ohkawa et al. [32], lipid peroxidation (LPO) in zebrafish liver homogenates was assessed by measuring the absorbance of reddish 2-thiobarbituric acid adducts at 532 nm using the molar extinction value of 1.56 × 10^5^ M^−1^ cm^−1^. The thiobarbituric acid reactive substances (TBARS) per gram of wet tissue mass were used to express the LPO values.

The concentration of protein carbonyls (PC) was determined by means of the reaction of 2,4-dinitrophenylhydrazine with carbonylated proteins in the sample of zebrafish liver homogenate [33]. Spectrophotometric detection of the resulting hydrazones solubilized in 8 M urea at 370 nm was used to reflect PC concentrations using the molar extinction coefficient of 2.2 × 10^4^ M^−1^ cm^−1^.

The soluble protein fraction, which was received by centrifuging the 10% (w:v) liver tissue homogenates in 20 mM HEPES-sucrose lysis buffer at 16,000 g for 45 min, was used to measure the level of reactive oxygen species (ROS). Using 485 nm excitation and 535 nm emission, the oxidative conversion of dihydrorhodamine 123 into fluorescent rhodamine 123 was seen [34]. The ROS value was expressed as relative fluorescence units (RFU) per g tissue.

DNA alkaline precipitation assay with Hoechst 33342 was used to assess the DNA strand breaks in hepatocytes [35]. The experiment was carried out in the presence of 0.4 M NaCl, 4 mM sodium cholate, and 0.1 M Tris (pH 9) to reduce the possibility of interference from sodium dodecyl sulphate traces [36]. The fluorescent product was read at 360 nm excitation and at 450 nm emission and the level of DNA strand breaks was expressed as the ratio of damaged DNA to their total level.

The nitric (II) oxide derivatives in the soluble protein fraction were monitored by NO level using Griess reagent [37]. In order to reduce NO_3_^−^ to NO_2_^−^ vanadium (III) chloride was used. The absorbance of resulted stable pinkish product was registered at 540 nm and the amount of nitrate/nitrite in the samples was recalculated using sodium nitrite as a standard.

### 2.4. Nonspecific Markers of Lysosomal and Hepatocellular Injury

A colorimetric assay based on the conversion of pyruvate into lactate in the presence of NADH was used to determine the activity of lactate dehydrogenase (LDH, EC 1.1.1.27) in the blood serum samples [38]. The LDH activity was determined based on the oxidation of NADH to NAD using molar extinction coefficient of 6220 M^−1^ cm^−1^.

The activity of cathepsin D was determined by measuring hemoglobin digestion at pH 3.2 [39]. The resulted product was monitored at 280 nm and the tyrosine was used as a standard to recalculate the total cathepsin D activity. Then, it was expressed as nmol tyrosine min^−1^ mg^−1^ of soluble protein.

### 2.5. Hormonal and Immune Markers

Cortisol (Cortisol Competitive ELISA Kit, Invitrogen, ThermoFisher, Waltham, MA, USA), Triiodothyronine (T3) (Triiodothyronine (T3) Competitive ELISA Kit, Invitrogen, ThermoFisher, Waltham, MA, USA), and serum IgM (Mouse IgM Elisa Kit, Sigma Aldrich, Burlington, MA, USA) concentrations in blood plasma were determined using 96-well-plate solid phase competitive ELISA Kits in accordance with the manufacturer’s protocols. Using serial dilutions of the samples and the standards, linearity for all ELISA assays was assessed and found to be acceptable (R^2^ = 0.97 ± 0.02).

### 2.6. Quantitative mRNA Expression Analysis

Phenol-based TRI Reagent^®^ (Sigma, St. Louis, MO, USA) was used to extract total RNA from the liver tissues in accordance with the manufacturer’s instructions. Potential DNA contaminations were removed from RNA samples using a TURBO DNA-free Kit (Thermo Fisher Scientific, Dreieich, Germany). cDNA was synthetized using a high-capacity cDNA Reverse Transcription Kit from 2 μg of the total RNA (Thermo Fisher Scientific, Dreieich, Germany). Using gene-specific primers, quantitative PCR was performed using a StepOnePlusTM Real-Time PCR System Thermal Cycling Block (Applied Biosystems, Thermo Fisher Scientific, Dreieich, Germany) and Biozym Blue S’Green qPCR Mix Separate ROX kit (Biozym Scientific GmbH, Hessisch Oldendorf, Germany) (Table 1). The reaction mixtures contained 10 μL of 2 × qPCR S’Green BlueMix and ROX additive mixture, 1.6 μL of each forward and reverse primer (to a final concentration of 0.4 μmol L^−1^), 4.8 μL PCR grade water and 2 μL cDNA sample. Tubulin was used as the housekeeping gene in order to normalize the expression of the target genes as the least variable across and within the experimental groups.

### 2.7. Statistical Data Processing

Having used the Kolmogorov–Smirnov and Levine tests, the normality of the data distribution and homogeneity of variances were examined. Box-Cox transformation was utilized when the data were not normally distributed or when the variances were diverse. The one-way ANOVA and the Tukey’s Honest Significant Difference (HSD) test were used to find out how experimental exposure affected the biological traits that were being studied. The Pearson correlation analysis was used to determine the relationship between biological traits.

To disclose the balance between antioxidants and prooxidants in zebrafish liver after exposure to studied pesticides, the integrative index of oxidative stress was calculated as IOS = (*TAC* + *CAT*)/(*ROS* + *TBARS* + *PC*) using standardized data [40]. Firstly, the deviation from the corresponding control (D(B)) was calculated for each of the selected biomarkers according to the formula D(B) = (Mean (B)_experimental_ − Mean (B)_control_)/Mean (B)_control_. Then, for D(B), each data point was increased by one. Finally, the ratio of deviated data for selected biomarkers was calculated and 2/3, computed as the ratio of cumulative quantity of antioxidants and prooxidants used in this case, and subtracted. In order to find out the total organism burden, the integrative index of biomarker response (IBR) was determined as the product of all studied traits after their standardization via the calculation of D(B).

Data were processed using Statistica 12.0 and Excel for Windows-2019. If the chance of Type I error (*p*) was less than 0.05, differences were deemed significant. Six biological replicates were employed for all biochemical and molecular analyses. All data were presented as means and standard errors of the mean (S.E.M.).

## 3. Results

### 3.1. Antioxidants

Terbuthylazine and malathion were able to induce TAC in the hepatocytes of zebrafish only at high concentrations, whereas pesticides at low concentrations or in combination with each other led to attenuation of TAC (Figure 1A).

Catalase was also suppressed by terbuthylazine in all tested concentrations and mixtures of pesticides. Malathion was the only exposure that did not affect CAT (Figure 1B).

The low concentration of terbuthylazine was the only exposure condition that led to an increase in total glutathione concentration (Figure 1C). Terbuthylazine at higher concentrations returned GSH to the control level. Malathion, on the other hand, decreased total glutathione at low concentrations but did not cause any changes at the higher studied concentrations. The total glutathione concentration in zebrafish liver was unaffected by co-exposure. Considering GSSG, both pesticides alone and particularly in combination caused a significant increase in GSSG (Figure 1D).

### 3.2. Prooxidant Events

Terbuthylazine exposure alone and in combination with malathion induced ROS production in the liver tissue of zebrafish (Figure 2A). Malathion in both studied concentrations had no impact on the ROS levels in the liver of zebrafish (Figure 2A). There was no concentration-dependent increase in ROS in response to terbuthylazine or conjoined exposures.

Almost all exposures provoked an increase in nitric oxide concentrations (Figure 2B). Only under malathion action at low concentration were no changes depicted. The value of NO changes did not depend on the magnitude of the terbuthylazine exposure.

The exposure to terbuthylazine and malathion led to significantly elevated liver tissue levels of TBARS. The effects of the terbuthylazine and malathion mixtures on the TBARS concentrations were more significant than those of the pesticides alone (Figure 2C). Terbuthylazine at low concentrations increased TBARS more significantly than at high concentrations.

Both studied pesticides at low concentrations caused the level of protein carbonyls in the liver of zebrafish to rise, while high concentrations of terbuthylazine and malathion had no discernible effect (Figure 2D). When pesticides were combined, they induced protein carbonylation without a clear concentration or pesticide-specificity direction.

### 3.3. Immune Response

Treatment of animals with terbuthylazine alone or in combination with malathion did not affect plasma IgM (*p* > 0.05) (Figure 3A). Exposure to malathion led to an increase in the IgM levels, and the concentration-dependence was inverse (Figure 3A).

Exposure to pesticides determined no changes in plasma T3 concentrations in all studied groups (*p* > 0.05) (Figure 3B).

Blood plasma cortisol remained silent against pesticide effects in all studied groups of fish (Figure 3C).

### 3.4. Mitochondrial and Lysosomal Response

Mitochondria isolated from fish liver showed signs of swelling, except for the ML group, in which no changes were found. The ability to cause an increase in mitochondrial swelling is enhanced in the range of malathion < terbuthylazine < pesticide mixture (Figure 3D).

SDH activity in the zebrafish liver was reduced significantly when it was exposed to the studied pesticides alone or when they were mixed together (Figure 3E).

Cathepsin D, a lysosomal protease, was found to increase after terbuthylazine exposure but was unaffected after malathion action. There was no evidence for the change in cathepsin D in the fish exposed to the pesticide mixtures (Figure 3F).

### 3.5. Signs of Geno- and Cytotoxicity

The level of DNA strand breaks was elevated in the liver of zebrafish exposed to terbuthylazine alone or to TH + ML mixture, as well as malathion at a low concentration (Figure 4A). The changes were more significant in the TL group than in the TH group. Exposure to the high concentration of malathion (50 μg L^−1^) had no effect on the DNA strand breaks level (Figure 4A). Only the TL + MH mixture led to a slight decrease in DNA fragmentation.

LDH activity was upregulated in a dose-dependent manner by terbuthylazine and malathion. The mixtures induced a two-fold increase in response when compared to pesticides effects alone (Figure 4B).

### 3.6. Expression of the Target Genes

In almost all studied cases, the expression of Nrf2 was noted to increase in the liver tissue of zebrafish. Exposure to the low concentration of terbuthylazine (2 μg L^−1^) and TL + MH mixture had no effect on the Nrf2 expression (Figure 5A).

Terbuthylazine exposure caused a significant activation of ubiquitination. There was no evidence for the change in ubiquitin expression in the liver of fish exposed to malathion alone or in combination with terbuthylazine (Figure 5B).

Exposure to pesticides led to elevated levels of caspase 3 in most studied groups (*p* < 0.05), except for the fish exposed to TH + ML (Figure 5C).

RAD51 was down-regulated by both the studied pesticides alone and in the co-exposed TH + ML group (by −35.3% ÷ −86.6%). The only increase in RAD51 expression (up to 170% relative to the control) was found in the fish exposed to TL + MH (Figure 5D).

There was no difference in BAX expression between control and pesticide-exposed fish, with one exception, detaching to the up-regulation of BAX after terbuthylazine exposure at the low studied concentration (Figure 5E). There was no evidence of Bcl-2 expression in the animals treated with pesticides (Figure 5F). The BAX/Bcl-2 ratio is reported to marginally increase in the liver of exposed zebrafish, except in the TH group.

### 3.7. Data Integration

In respect to the integrative oxidative stress index calculation, all treatments induced prooxidative changes in zebrafish hepatocytes (IOS < 0) (Figure 6). The deepest differences were noted after co-joint exposure. According to the integrative biomarker response index, terbuthylazine caused more significant variability in the studied indices at lower concentrations than malathion did at higher concentrations. The highest variability and engagement of biochemical pathways were observed in the TTH + ML group (Figure 6).

## 4. Discussion

Oxidative stress might be induced by breaking the balance between oxidative lesions, namely lipid peroxides and protein carbonyls, and a decrease in antioxidant capacity [41]. Some studies have suggested that pesticides affect living organisms by enhancing ROS production, which, in turn, causes oxidative damage to biomolecules [42]. Herein, we have shown that malathion and particularly terbuthylazine were able to induce reactive oxygen and nitrogen species derivation, suppress antioxidants, and induce oxidative injury in the liver of treated zebrafish, which corroborates with the early findings that they both can perform as oxidants and react with reducing agents [43,44]. Although the depths of prooxidant changes were more significant in animals exposed to low environmental realistic concentrations of terbuthylazine and malathion, an imbalance was observed after pesticide exposure to high concentrations and their mixture accorded with more profound LDH leakage and mitochondrial swelling as the markers of tissue damage. It is highly likely that the accumulation of ROS and oxidative damage products may well activate stress-induced transcription factors and the production of proinflammatory and anti-inflammatory cytokines, following the deterioration of stress response and appearance of histopathological signs [45]. Nonetheless, being subjected to the prolonged negative impact of pesticides, an organism can suffer from irreversible tissue damage and acute health effects, particularly when defensive systems reach their tolerance limits.

Terbuthylazine is known to produce redox active metabolites, including terbuthylazine-desethyl and terbuthylazine-2-hydroxy [43], which in turn caused lipid peroxidation and alterations in antioxidants in red swamp crayfish (*Procambarus clarkii*) and common carp (*Cyprinus carpio*) [46,47]. Although the concentrations of terbuthylazine we used for animal exposition were much lower than previously studied, terbuthylazine alone and in mixture was able to induce ROS and nitric oxide in the liver of zebrafish. In contrast, malathion did not tend to stimulate radical production but downregulated antioxidants more profoundly than terbuthylazine. It is highly likely that terbuthylazine and malathion trigger different oxidative stress-sensitive pathways, which in the case of conjoined exposure calls for more profound oxidative stress than each of the exposures alone.

In aquatic animals, malondialdehyde and protein carbonyls, as the products of lipid and protein oxidative damage, have received the greatest attention in the study of toxic effects due to their wide exploration, simplicity of determination, and sensitivity [5,41,48]. It is worth mentioning that TBARS levels in the liver, kidney, and gills of fish increased after short- and long-term exposure to numerous pesticides, including atrazine (30–90 µg L^−1^), chlorpyrifos (1.4 and 2.44 µg L^−1^), cypermethrin (0.5 or 1 µg L^−1^), glyphosate (15 µg L^−1^ and 500 µg L^−1^), and methyl parathion (0.009–0.09 ppm) [20,49,50,51,52]. While reactions to the effects of organophosphate pesticides have been fairly well studied, the effects of chloro-s-triazine have been almost completely overlooked [5]. Only some examinations exist regarding the induction of lipid peroxidation in yellow-tailed tetra fish (*Astyanax altiparanae*) after atrazine exposure [53]. Our data showed that terbuthylazine moderately induces lipid peroxidation and protein carbonylation in zebrafish liver, particularly at environmentally realistic concentrations. However, when combined with malathion, they potentiate each other, and oxidative damage to lipids and proteins increases significantly. The late event correlated with LDH leakage as a cytotoxic sign (r > 0.63, *p* < 0.001) (Appendix A).

Glutathione is reported to play an important role in maintaining cellular homeostasis by scavenging ROS generated from metabolic processes and external sources. The disruption of the cellular redox pool regulated by GSH leads not only to oxidative stress but also to the endogenous overflow of NO, which in turn facilitates protein nitration, S-nitrosylation, and DNA strand breaks [54]. Low concentrations of terbuthylazine, according to our findings, activate glutathione synthesis, which partly compensates the decline of catalase activity, supports the cellular redox state, scavenges NO, and mitigates the oxidative injury and cytotoxicity induced by the pesticide effect. However, as the concentration of terbuthylazine in the environment increased, the anti-radical capacity of GSH in zebrafish livers was significantly decreased. Although the level of oxidative lesions in the TH group decreased when compared with the TL group, cell injury became more progressive, which revealed higher LDH leakage and mitochondrial swelling. Obviously, GSH in zebrafish serves as an important buffer for NO, which may well perform as the key mediator of cell harm in terbuthylazine toxicity and holds the organism back from excessive toxicity. The important role of NO was proved also for terbuthylazine-induced endocrine disruption in human ovarian granulosa cells [55].

Considering how easily mitochondria can be harmed, they have to be extremely vulnerable to environmental toxins [56,57]. Both malathion and terbuthylazine, in the form of its metabolite, 2-hydroxyterbuthylazine, have lipophilic properties and, therefore, can enter organisms by passive diffusion and then assemble inside the mitochondrial matrix due to the difference in charge between the matrix and the cytoplasm. Mitochondrial reactive oxygen species, which are expected to be generated in the response to pesticides, can obviously damage the electron transport chain, affect membrane permeability, and cause mutations in mitochondrial DNA [19,20,57]. The last event is the riskiest, in view of the fact that compared to the nucleus, mitochondria have a lower potential for DNA repair [58]. Our findings proved the ability of malathion and terbuthylazine, singly and, particularly, in mixture, to significantly affect zebrafish mitochondria, as evaluated by mitochondrial swelling and complex II of the electron transport chain. All of these events have the potential to alter the energetic budget as well as affect the expression of cell cycle and stress-related genes, which have been linked in mammals [59,60].

Since Nrf2 is a true pleiotropic transcription factor, it is thought to be a master-regulator of the expression of genes involved in detoxification, inflammation, and the oxidative stress response, all of which contribute to cellular defense against toxic effects and oxidative stress and make Nrf2 a potent modulator of lifespan [61]. The effective regulation of Nrf2 signaling is believed to support the cellular redox state and protect cells from toxin-induced apoptosis and necrosis [62,63]. In particular, Nrf2 pathway activation can effectively inhibit cell inflammation and the expression of receptor-interacting proteins RIP3, which play a vital role in the regulation of the necrosis signaling pathway in palmitate-stimulated mouse cells [62]. In the present study, the expression of Nrf2 in the liver of zebrafish agreed with cathepsin D activity and the expression of ubiquitin, both implicated in apoptosis/necrosis [64] and correlated each other (r = 0.73, *p* < 0.001) (Appendix A). As ROS and RNS may affect the ubiquitin proteasome system that degrades misfolded proteins [22], overexpression of Nrf2 after malathion and mixture exposure was highly likely to attenuate the negative impact of pesticides on fish and restrain zebrafish cells and tissues from deep injury and promotion of necrosis.

In fish, innate immune cells and adaptive immune cells such as leukocytes and natural killer cells both play a role in the immunological response to stressors. In particular, it was observed that acute exposure to organophosphate pesticides in cyprinids and cichlidaes provoked a decrease in IgM, phagocytic activity, but enhanced the expression of IL-6, IL-8, and TNF-α [65,66,67]. Meanwhile, data on the effects of triazine herbicides are limited and describe mainly morphological and functional changes in immunocompetent cells [68]. According to our study, zebrafish exposed to malathion showed higher levels of IgM, particularly in the case of the ML-group. As malathion levels in the environment have increased, so has its potential to inhibit innate immunity in zebrafish, as evidenced by significantly lower levels of IgM in the MH group compared to the ML group, as well as humoral immune response suppression after acute malathion exposure in fish [69]. All of that can attenuate the ability of fish to withstand viral and bacterial diseases and then pave the way to biodiversity loss. Terbuthylazine, on the other hand, appears to be immune-indifferent for fish in realistic environmental concentrations because it did not affect immunoglobulin synthesis, though acute exposure may result in harmful outcomes [68].

Existing findings suggest that organophosphates and triazine pesticides, which are known to be endocrine disruptors, may influence the corticosteroid levels or thyroid equilibrium in fish [5]. Our study showed no significant changes in either thyrotropin or cortisol in exposed zebrafish. Being involved in gluconeogenesis, steady-state cortisol may indicate no significant disorders of carbohydrate metabolism. Additionally, cortisol can be manipulated by the adreno-corticotropic hormone, which in turn can be promoted by acetylcholine [70]. An increase in acetylcholine levels has been linked to organophosphate exposure [71]. So, no changes in cortisol can point indirectly to no alterations in the cholinergic system, and there is no specific neurotoxicity of malathion and terbuthylazine in studied concentrations in zebrafish.

Acute stress-related changes in the hypothalamic-pituitary-thyroid axis are unique to non-thyroid systemic impairments. It is known as “non-thyroid impairment syndrome” and primarily manifests as a drop in triiodothyronine levels. This debilitating condition develops quickly in hazardous environments and is associated with a poor prognosis in higher vertebrates, including humans [72]. Considering these data, we are able to conclude that the exposure to low, environmentally realistic concentrations of malathion and terbuthylazine alone and in combination was insufficient to cause significant alterations in the hypothalamic-pituitary-thyroid axis in zebrafish and provoke acute detrimental effects.

Zebrafish were affected by both terbuthylazine and malathion, so that cytotoxicity evaluated by LDH leakage, DNA strand breaks, and cathepsin D were observed in the fish liver after exposure to all tested concentrations. Our findings are consistent with previous research that found organophosphate pesticides and atrazine, a triazine pesticide, to be moderate or even highly toxic to fish [5,73]. When zebrafish were handled with the mixture of pesticides, LDH leakage was multiplied but, in contrast, DNA strand breaks were decreased, particularly in the TL + MH group. When compared to a single exposure, it is likely enough that as the exposure burden increases, DNA-repair mechanisms will begin to heal the damage, resulting in a significant decrease in DNA strand breaks. This assumption is supported by the significantly higher expression of RAD51 and the multiple factors involved in faithful DNA replication, repair, and recombination, as well as in the innate immune response [74], in TL + MH exposed animals. In the opposite direction, higher levels of DNA strand breaks correlated with a low level of RAD 51 (r = −0.41, *p* = 0.006) (Appendix A).

Understanding the cumulative toxicological effects of pesticide combinations, which are common in the environment, is crucial for wildlife conservation. However, it is difficult to examine all potential combinations of chemicals, exposure patterns, and complicated interactions, and there is a dearth of experimental data sets regarding the toxicity of mixtures [75]. It is expected that malathion does not react with chloro-s-triazines, so a combinatorial effect can be explained only by increasing the concentration burden. Pesticides most likely activated different targets in zebrafish in our case study, and only cytotoxicity markers, such as LDH leakage, mitochondrial swelling, and lipid peroxidation, reflected cumulative toxicity. The sensitivity of TBARS to the combined exposure in fish was also reported earlier [20,76], and can therefore be proposed as a candidate for non-specific cumulative risk assessment.

## 5. Conclusions

To sum up, our experiment demonstrates that both terbuthylazine and malathion, singly and/or in combination, induce oxidative stress, ROS, and RNS in zebrafish liver, highly likely triggering different oxidative stress-activated signaling pathways. The toxicity of the studied pesticides and their mixtures to zebrafish decreased in the order: mix of terbuthylazine and malathion > terbuthylazine > malathion. Terbuthylazine disturbed the antioxidant-prooxidant cellular balance more significantly than malathion and induced DNA fragmentation and lysosomal destabilization. Malathion, in turn, was able to cause the immune response observed by IgM. The common adverse effects of studied pesticides included lipid peroxidation, mitochondrial swelling, and LDH leakage. Meanwhile, malathion and terbuthylazine in environmentally relevant concentrations were proven not to cause significant alterations in the hypothalamic-pituitary-thyroid axis in zebrafish. While we have made some progress in our understanding of pesticide toxicity, much more research is needed to fully understand the molecular pathways by which pesticides, particularly chloro-s-triazines, can affect fish and other aquatic animals.

## Figures and Tables

**Figure 1 animals-13-01029-f001:**
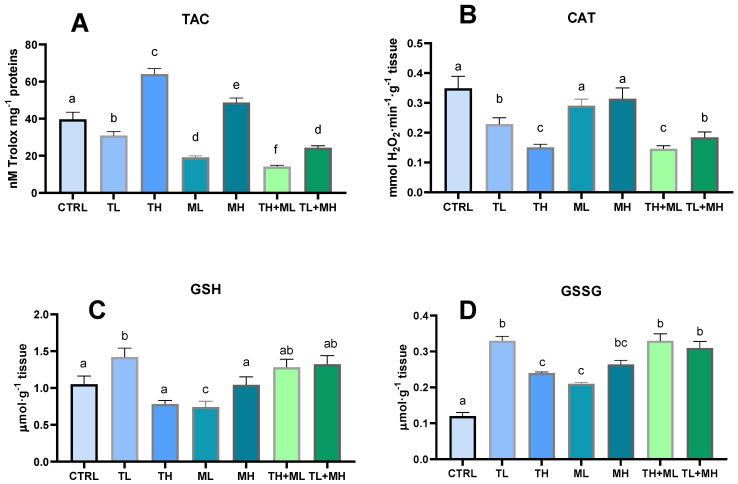
Effects of terbuthylazine (2 µg L^−1^ (TL) and 30 µg L^−1^ (TH)) and malathion (5 µg L^−1^ (ML) and 50 µg L^−1^ (MH)) exposures on antioxidants status in the liver of zebrafish. (**A**) Total antioxidant capacity; (**B**) catalase; (**C**) Glutathione total; (**D**) Glutathione oxidized. When columns share the different letters (a–f), they represent significantly different values (*p* < 0.05). The means and the standard errors of the mean are presented. *N* = 6.

**Figure 2 animals-13-01029-f002:**
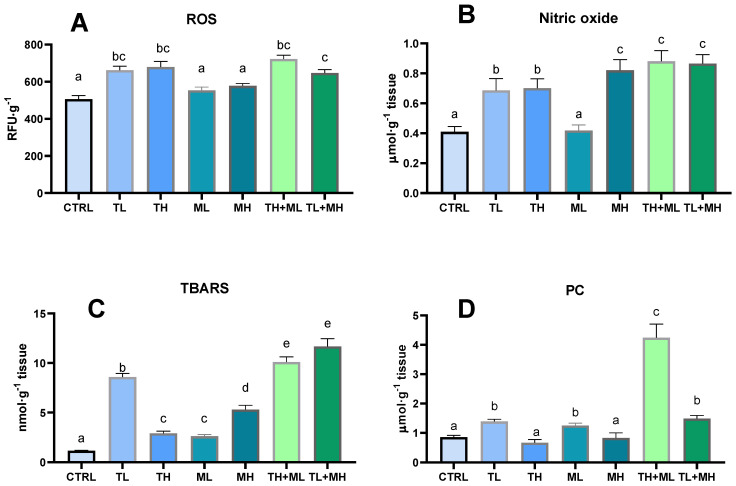
Effects of terbuthylazine (2 µg L^−1^ (TL) and 30 µg L^−1^ (TH)) and malathion (5 µg L^−1^ (ML) and 50 µg L^−1^ (MH)) exposures on reactive oxygen and nitrogen species as well as oxidative lesions in the liver of zebrafish. (**A**) Reactive oxygen species; (**B**) Nitric oxide; (**C**) TBA-reactive substances; **D**, Protein carbonyls. When columns share the different letters (a–e), they represent significantly different values (*p* < 0.05). The means and the standard errors of the mean are presented. *N* = 6.

**Figure 3 animals-13-01029-f003:**
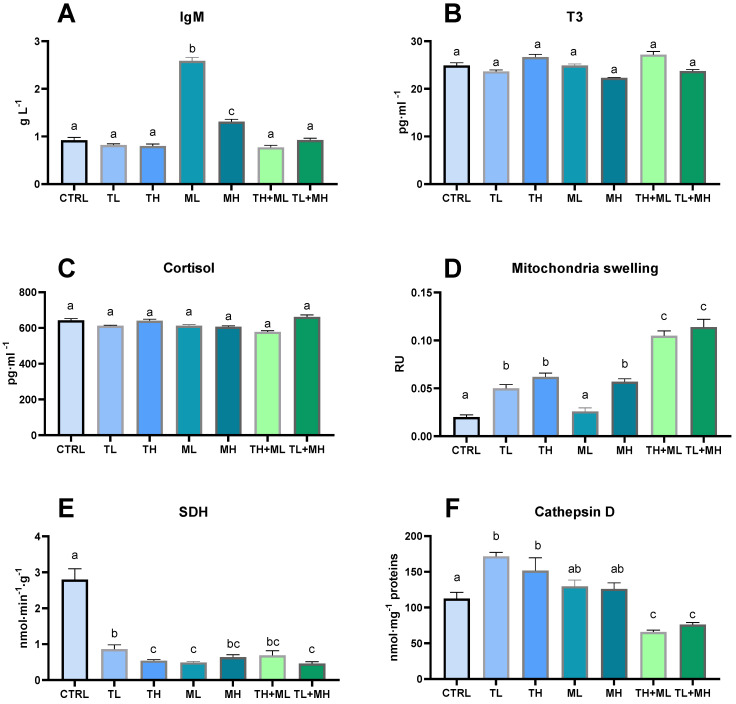
Effects of terbuthylazine (2 µg L^−1^ (TL) and 30 µg L^−1^ (TH)) and malathion (5 µg L^−1^ (ML) and 50 µg L^1^ (MH)) exposures on immune (**A**) and endocrine (**B**,**C**) traits in zebrafish blood and mitochondrial (**D**,**E**) and lysosome (**F**) related indices in hepatocytes. Columns with different letters have significantly different values (*p* < 0.05). The means and the standard errors of the mean are presented. *N* = 6.

**Figure 4 animals-13-01029-f004:**
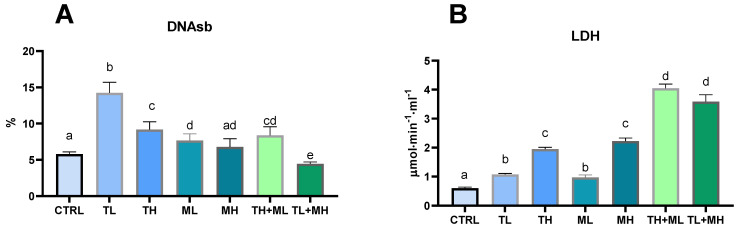
Effects of terbuthylazine (2 µg L^−1^ (TL) and 30 µg L^−1^ (TH)) and malathion (5 µg L^−1^ (ML) and 50 µg L^−1^ (MH)) exposures on the ratio of DNA strand breaks in hepatocytes and lactate dehydrogenase leakage in blood of zebrafish. (**A**) DNA strand breaks; (**B**) Lactate dehydrogenase. Columns with different letters (a–e) have significantly different values (*p* < 0.05). The means and the standard errors of the mean are presented. *N* = 6.

**Figure 5 animals-13-01029-f005:**
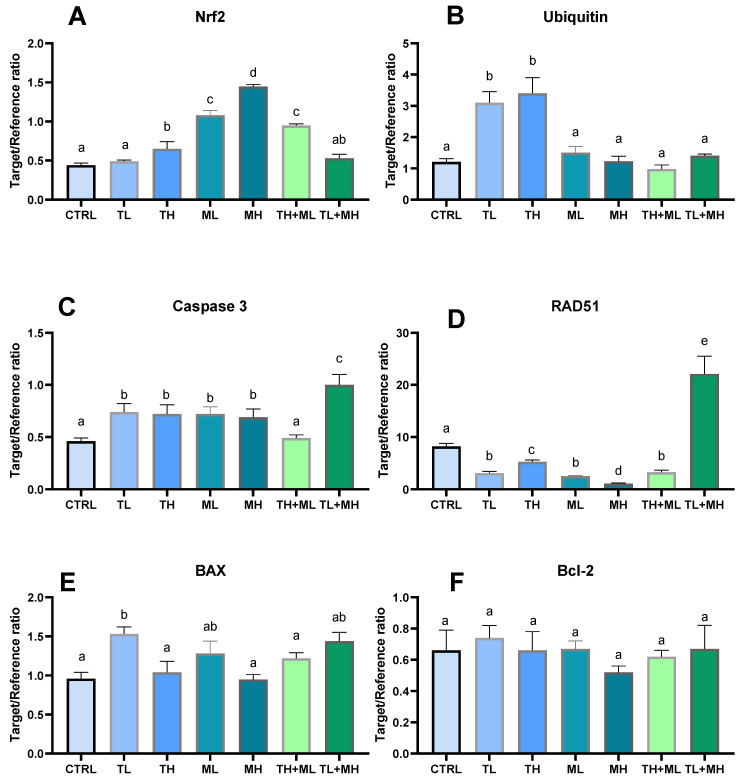
Effects of terbuthylazine (2 µg L^−1^ (TL) and 30 µg L^−1^ (TH)) and malathion (5 µg L^−1^ (ML) and 50 µg L^−1^ (MH)) exposures on mRNA expression of apoptotic, oxidative stress, and DNA damage and repair markers in the liver of zebrafish. (**A**) the nuclear factor erythroid 2–related factor 2, (**B**) Ubiquitin; (**C**) Caspase 3; (**D**) DNA repair protein RAD51; (**E**) B-cell lymphoma protein 2 (Bcl-2)-associated X; (**F**) B-cell lymphoma protein 2. Columns with different letters have significantly different values (*p* < 0.05). The means and the standard errors of the mean are presented. *N* = 6.

**Figure 6 animals-13-01029-f006:**
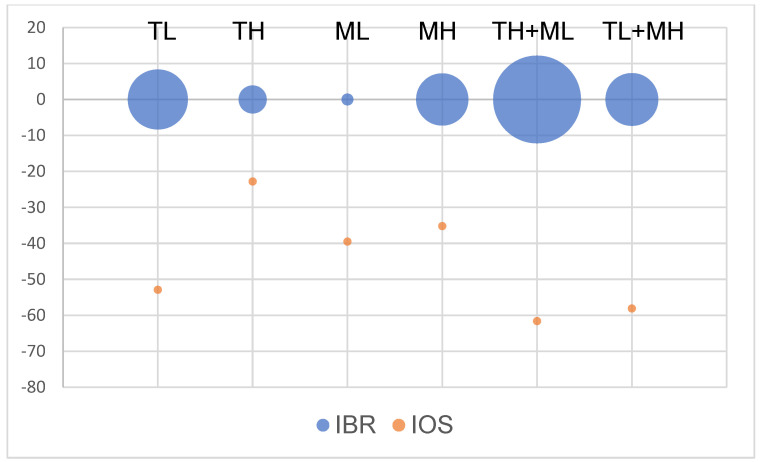
Integrative oxidative stress response and biomarker response in zebrafish after the effects of terbuthylazine (2 µg L^−1^ (TL) and 30 µg L^−1^ (TH)) and malathion (5 µg L^−1^ (ML) and 50 µg L^−1^ (MH)). IBR- integrative index of biomarker response, IOS – integrative index of oxidative stress.

**Table 1 animals-13-01029-t001:** Gene-specific primers for the studied target and housekeeping genes from *D. rerio*. Gene abbreviations: Bcl-2—B-cell lymphoma 2; BAX—Bcl-2-associated X protein; Nrf2—nuclear factor erythroid 2-related factor 2; Vtg—Vitellogenin.

Gene	Primers	Primer (5′–3′)	Tm	Ta	NCBI Accession No.
Caspase 3a	Forward	GACGCAAAGCGTGTGGATAC	58.4	58.4	NM_131877.3
	Reverse	GCCGATGTTGGGGTAGTTCA	59.6		
Bcl-2	Forward	GCGGAGGGAACAACTCTGAA	59.8	59.8	AY695820.1
	Reverse	ATCCCGTAACACCCGGTAGA	60.0		
BAX	Forward	GACTTGGGAGCTGCACTTCT	59.8	59.8	NM_131562.2
	Reverse	CTGACTCCGGGTCACTTCAG	60.1		
Nrf2	Forward	CGGGAGATTTCAGCTCAGGG	60.4	59.8	NM_182889.1
	Reverse	CGAAGGATCCGTCTTCGGTT	59.8		
RAD51	Forward	CCGCTATGATGACCGAGTCC	60.0	59.9	NM_213206.2
	Reverse	CAGCATACGCAGAAAGCGTC	59.9		
Ubiquitin	Forward	GTCTCCGAGGAGGCTCAGAT	60.4	59.8	NM_001013272.2
	Reverse	GTGAGTGCATAAGCAGGGGA	59.8		
Tubulin	Forward	GTTGGAGCTGAGAGTGTGGAA	59.9	59.0	NM_194388.2
	Reverse	CAATGGACAGGAAACACAGCA	59.0		

## Data Availability

The datasets generated during and/or analysed during the current study are available from the corresponding author on reasonable request.

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
