# Peer review of "Molecular and Biochemical Evidence of the Toxic Effects of Terbuthylazine and Malathion in Zebrafish"

_animals, 2023, doi:10.3390/ani13061029_

Round 1

Reviewer 1 Report

Authors did a really good work. They summarized a lot of data into interesting and useful manuscript on topic that is of importance for readers of Animals. I just have some minor suggestion.

line 19 delete "in zebrafish"

keywords may differ from those in the title, for example Danio rerio instead of zebra, or organophosphate insecticide instead of malathion

I noticed that paragraphs in Introduction start with Although or Due to, so  maybe for line 94 you can start sentence "Movement and relatively..., make fish..."

lines 64-70 do not seem too important for ms, maybe reduce them or delete.

In experimental exposure line 123 you mention "225 specimens" and later in lines 136-137 "Three replicate tanks with 15 fish each for each experimental treatment." It means 3x15x (6 treatments + control)= 305. Just clarify this please

Can you add references based on which you calculated IOS and IBR and how you standardize data.

In discussion you often use "Highly likely" maybe change it

line 380 "as" instead "correlated with" as you did not calculate correlation

I also noticed that you gave correlations for LDH, cathepsin D and ubiquitin, ... (lines 453, 412, 505). You should add them in result section and provide the rest of correlations in Supplementary material

Explanations why you did not get any differences in cortisol and HPT axis are redundant, you can remove them as you  already have long enough discussion.

Reviewer 2 Report

I advise against the publication of the manuscript with ID (animals-2248392) in Animals. Three main reasons have led me to this decision.

The first reason: -

Lines 131-136: Although the authors have illustrated that they tested the worst-case scenarios by exposure of zebrafish to pesticides higher than those reported as relevant environmental values in freshwaters, I found that authors must conduct a preliminary exposure test to determine the LC50 (lethal concentration which will kill 50% of the exposed fish). This is important to use an exposure dose below the LC50 and to determine its effects on fish. According to Weltje and Sumpter (2017), it was confirmed that the primary reason for conducting ecotoxicology research is to obtain information that can be used by regulators to protect the environment from the many chemicals accidentally or intentionally released into the environment.

Weltje, L. and Sumpter, J. P. (2017). What makes a concentration environmentally relevant? Critique and a proposal. Environ. Sci. Technol. 51, 20, 11520–11521

The second reason: -

I found that One-Way ANOVA is incorrect analysis for your exposure doses, even if the exposure doses were correctly chosen. Please go to Lines 127-131. You chose 6 experimental groups that divided into the following items:-

A) Single exposures

1) Terbuthylazine, 2 μg L−1 (TL), 2) Terbuthylazine, 30 μg L−1 (TH),

3) Malathion, 5 μg L−1 (ML), 4) Malathion, 50 μg L−1 (MH),

B) Combined exposures

5) Terbuthylazine 30 μg L−1 and malathion 5 μg L−1 130 (TH+ML),

6) Terbuthylazine 2 μg L−1 and malathion 50 μg L−1 (TL+MH).

One way ANOVA is correctly conducted for Single exposures ONLY.

For combined exposures, you MUST conduct Two-Way ANOVA. This is important to elucidate the effects of either doses of the tested pesticide (Terbuthylazine and Malathion) or their interactions (insecticide type and dose) in the combinations. This will help you to determine if these combinations made additive, synergism, or antagonistic interactions on the examined fish species.

The third reason: -

This is a technical one as the authors in Lines 144-146 did not describe how they maintain the exposure doses of the tested pesticides after water renewal.

Author Response

‘I advise against the publication of the manuscript with ID (animals-2248392) in Animals. Three main reasons have led me to this decision.

The first reason: -

Lines 131-136: Although the authors have illustrated that they tested the worst-case scenarios by exposure of zebrafish to pesticides higher than those reported as relevant environmental values in freshwaters, I found that authors must conduct a preliminary exposure test to determine the LC50 (lethal concentration which will kill 50% of the exposed fish). This is important to use an exposure dose below the LC50 and to determine its effects on fish. According to Weltje and Sumpter (2017), it was confirmed that the primary reason for conducting ecotoxicology research is to obtain information that can be used by regulators to protect the environment from the many chemicals accidentally or intentionally released into the environment.

Weltje, L. and Sumpter, J. P. (2017). What makes a concentration environmentally relevant? Critique and a proposal.‏ Environ. Sci. Technol. 51, 20, 11520–11521

Authors. We wish to thank you for your profound analysis of our work. Let me provide an explanation of the first reason. Lethal concentration (LC50) estimation is a widely used measure of toxicity and is used extensively in ecological risk assessment. Therefore, these data have been previously determined for terbuthylazine and particularly for malathion. As an example, the LC50 for malathion in zebrafish is 5.29 mg·L-1 (http://www.ijesd.org/vol8/1043-D729.pdf) and for terbuthylazine in carp embryos is 441.6μg/L (LC5031 days). However, the lethal concentration (LC50) has recently been heavily criticized as a toxicological endpoint for use in ecological risk assessment (https://doi.org/10.1897/IEAM_2004-002r.1). On the one hand, the environmental concentrations of xenobiotics, among them studied pesticides, are much lower than LC50, and it is far-fetched that organisms may well face such acute concentrations. The LC50 approach, on the other hand, needs a lot of animal specimens, which has been the platform of the discussion in different bioethics rounds (https://www.ecetoc.org/wp-content/uploads/2014/08/MON-006.pdf) which advised against this approach. Despite the fact that the studied concentrations were much lower than the LC50, they caused significant changes in biochemical pathways, which obviously resulted in numerous negative outcomes and may disrupt metabolic and fitness costs. It supports the relevance of the present investigation. 

The second reason: -

I found that One-Way ANOVA is incorrect analysis for your exposure doses, even if the exposure doses were correctly chosen. Please go to Lines 127-131. You chose 6 experimental groups that divided into the following items:-

    A) Single exposures

1) Terbuthylazine, 2 μg L−1 (TL), 2) Terbuthylazine, 30 μg L−1 (TH),

3) Malathion, 5 μg L−1 (ML), 4) Malathion, 50 μg L−1 (MH),

    B) Combined exposures

5) Terbuthylazine 30 μg L−1 and malathion 5 μg L−1 130 (TH+ML),

6) Terbuthylazine 2 μg L−1 and malathion 50 μg L−1 (TL+MH).

One way ANOVA is correctly conducted for Single exposures ONLY.

For combined exposures, you MUST conduct Two-Way ANOVA. This is important to elucidate the effects of either doses of the tested pesticide (Terbuthylazine and Malathion) or their interactions (insecticide type and dose) in the combinations. This will help you to determine if these combinations made additive, synergism, or antagonistic interactions on the examined fish species.

Authors: Thank you for your comment. We used a one-way ANOVA in order to test if at least one of the treatment means is significantly different from the others. The categorical independent variable included each studied condition separately, so we had seven groups in total. In general, the two-way ANOVA examines the influence of two different categorical independent variables on one continuous dependent variable. Having multiple effects and, therefore, categorical variables (pesticide type (herbicide/insecticide and their mixture), concentration (low/high/varieties of their combinations), and mode of exposure (single or mixed exposure), we have been inclined to use integral index calculations and their size effect (integral index of oxidative stress and biomarker responses) to disclose overall body burden instead of two-way ANOVA.

The third reason: -

This is a technical one as the authors in Lines 144-146 did not describe how they maintain the exposure doses of the tested pesticides after water renewal.

Authors: Thank you so much. Due to your valuable comment the correspondent information has been added to the text.

Reviewer 3 Report

The manuscript "Molecular and Biochemical Evidence of the Toxic Effects of 2 Terbuthylazine and Malathion in Zebrafish" by Khatib and colleagues, is certainly interesting and well structured, the analyses performed are sound and the results well presented. Nevertheless, it needs revision in the materials and methods section.

-Line 125-159: The authors stated :"225 specimens of zebrafish of the same size (3.4 ± 0.4 cm) and the same mass (0.9 ± 0.2 g) were purchased from a local vendor  (Rubkavdoma, Kyiv, UA). The fish were randomly assigned to one control group (not exposed to chemicals) and six experimental groups, each of which was exposed to terbuthylazine, malathion, or mixtures in two different concentrations". In the followings lines, the authors also stated: "The control fish were kept in filtered tap water for 14 days. Three replicate tanks with 15 fish each in 10 L of water were utilized for each experimental treatment".It's not clear how many fish the authors used for this exposure because if we assume to have 225 fish with 15 fish per group (6 plus 1 control) and 3 replicates for each "treatment" group ,all of this study require 285 fish at least.Please explain the exact number of fish used for all the analysis and revise the text accordingly.To improve the quality of the manuscript i suggest the authors to take a clue from this study https://doi.org/10.3390/toxics10050272 and cite it.

Author Response

The manuscript "Molecular and Biochemical Evidence of the Toxic Effects of 2 Terbuthylazine and Malathion in Zebrafish" by Khatib and colleagues, is certainly interesting and well structured, the analyses performed are sound and the results well presented. Nevertheless, it needs revision in the materials and methods section.

Authors: Thank you very much for your review, a positive feedback on our work, and additional comments which we addressed below while revising our manuscript.

-Line 125-159: The authors stated :"225 specimens of zebrafish of the same size (3.4 ± 0.4 cm) and the same mass (0.9 ± 0.2 g) were purchased from a local vendor  (Rubkavdoma, Kyiv, UA). The fish were randomly assigned to one control group (not exposed to chemicals) and six experimental groups, each of which was exposed to terbuthylazine, malathion, or mixtures in two different concentrations". In the followings lines, the authors also stated: "The control fish were kept in filtered tap water for 14 days. Three replicate tanks with 15 fish each in 10 L of water were utilized for each experimental treatment".It's not clear how many fish the authors used for this exposure because if we assume to have 225 fish with 15 fish per group (6 plus 1 control) and 3 replicates for each "treatment" group ,all of this study require 285 fish at least.Please explain the exact number of fish used for all the analysis and revise the text accordingly.To improve the quality of the manuscript i suggest the authors to take a clue from this study https://doi.org/10.3390/toxics10050272 and cite it.

Authors: Thank you for your comment, which helped improve a technical mistake. You were right: the total number of specimens were 3 replicates´15 specimen´7 groups = 315 and the correspondent correction was settled in the text. The manuscript you recommended was definitely helpful in improving the quality of the material and methods paragraph, and we will cite it the very next time.

Round 2

Reviewer 2 Report

Authors have properly addressed the comments raised by the anonymous reviewer. However, there are several minor comments. The following comments should be done by the authors before the manuscript considered for publication in Animals. I noticed that the authors used NCBI GenBank accession numbers that related to other researchers and the authors did not design them by themselves. In this case, the authors should add the following information in Table 1.

1. Gene efficiency %.

2. Tm and Annealing temperatures.

3. References: From where you get these primers.

(NM_131877.3)

Iyer, H., Shen, K., Meireles, A. M., & Talbot, W. S. (2022). A lysosomal regulatory circuit essential for the development and function of microglia. Science Advances, 8(35), eabp8321.

(AY695820.1)

Langenau, D. M., Jette, C., Berghmans, S., Palomero, T., Kanki, J. P., Kutok, J. L., & Look, A. T. (2005). Suppression of apoptosis by bcl-2 overexpression in lymphoid cells of transgenic zebrafish. Blood, 105(8), 3278-3285.

(NM_131562.2)

Chen, J., Zhang, M., Zou, H., Aniagu, S., Jiang, Y., & Chen, T. (2022). Synergistic protective effects of folic acid and resveratrol against fine particulate matter-induced heart malformations in zebrafish embryos. Ecotoxicology and Environmental Safety, 241, 113825.

(NM_182889.1)

Jin, B., Xie, L., Zhan, D., Zhou, L., Feng, Z., He, J., ... & Li, L. (2022). Nrf2 dictates the neuronal survival and differentiation of embryonic zebrafish harboring compromised alanyl-tRNA synthetase. Development, 149(17), dev200342.

(NM_213206.2)

Rozenblat, R., Tovin, A., Zada, D., Lebenthal-Loinger, I., Lerer-Goldshtein, T., & Appelbaum, L. (2022). Genetic and Neurological Deficiencies in the Visual System of mct8 Mutant Zebrafish. International Journal of Molecular Sciences, 23(5), 2464.

(NM_001013272.2)

Schultz‐Rogers, L. E., Thayer, M. L., Kambakam, S., Wierson, W. A., Helmer, J. A., Wishman, M. D., ... & McGrail, M. (2022). Rbbp4 loss disrupts neural progenitor cell cycle regulation independent of Rb and leads to Tp53 acetylation and apoptosis. Developmental Dynamics, 251(8), 1267-1290.

(NM_194388.2)

Whyte-Fagundes, P., Taskina, D., Safarian, N., Zoidl, C., Carlen, P. L., Donaldson, L. W., & Zoidl, G. R. (2022). Panx1 channels promote both anti-and pro-seizure-like activities in the zebrafish via p2rx7 receptors and ATP signaling. Communications Biology, 5(1), 472.

This information is very important for researchers for future replication of your findings. In addition to give the rights to the previously published papers.

Author Response

Authors have properly addressed the comments raised by the anonymous reviewer. However, there are several minor comments. The following comments should be done by the authors before the manuscript considered for publication in Animals. I noticed that the authors used NCBI GenBank accession numbers that related to other researchers and the authors did not design them by themselves. In this case, the authors should add the following information in Table 1.

  1. Gene efficiency %.
  2. Tm and Annealing temperatures.
  3. References: From where you get these primers.

Authors: Thank you very much for your review, a positive feedback on our work, and additional comments which we addressed below while revising our manuscript.

Tm and Annealing temperatures were added to Table 1. Meanwhile, I wish I would not add references, as the common mode to show primers includes only forward and reverse primers and the NCBI accession number, which is an open-source database. Frankly speaking, it is hard to recognize which reference from several listed in NCBI we have to cite. As an example,

(NM_194388.2)

"Whyte-Fagundes, P., Taskina, D., Safarian, N., Zoidl, C., Carlen, P. L., Donaldson, L. W., & Zoidl, G. R. (2022). Panx1 channels promote both anti-and pro-seizure-like activities in the zebrafish via p2rx7 receptors and ATP signaling. Communications Biology, 5(1), 472." Is only one of the 10 referred to on the correspondent page https://www.ncbi.nlm.nih.gov/nuccore/NM_194388.2